# Smoothing the Surface and Improving the Electrochemical Properties of Na*_x_*MnO_2_ by a Wet Chemical Method

**DOI:** 10.3390/nano10020246

**Published:** 2020-01-30

**Authors:** Siliang Zhao, Zhiping Lin, Fugen Wu, Feng Xiao, Jiantie Xu

**Affiliations:** 1School of materials and energy, Guangdong University of Technology, Guangzhou 510006, China; Liang88552@163.com; 2School of Physics and optoelectronic engineering, Guangdong University of Technology, Guangzhou 510006, China; 3Guangdong Provincial Key Laboratory of Atmospheric Environment and Pollution Control, School of Environment and Energy, South China University of Technology, Guangzhou 510640, China; fengxiao@scut.edu.cn (F.X.); jiantiexu@scut.edu.cn (J.X.)

**Keywords:** Na*_x_*MnO_2_ (NMO), wet chemical method, surface, electrochemical properties

## Abstract

Na*_x_*MnO_2_ (NMO) is treated by a wet chemical method in this paper. The treated NMO can form a copper oxide coating layer, and some of the coating layer can be peeled off, smoothing the surface of particles. Electrochemical measurement shows that treated NMO can maintain 72.6% of its specific capacity after 300 cycles, which is better than the 58.7% specific capacity of untreated NMO materials. Additionally, the ratio of capacity remaining rate can be improved from an initial 87% to 99.5%. So, this wet chemical method is available to smooth the electrode surface and reduce the internal impedance, and thus to effectively improve electrochemical performance during the battery cycle.

## 1. Introduction

Along with the rapid development of science and technology, nothing in daily life can be separated from energies. The development of energy storage and its rational utilization is a main melody of science and technology development. In 1991, the SONY company launched the first commercial lithium ion battery; since then, the lithium ion battery has been more and more widely used in daily life. However, we must be prepared for the exhaustion of Li resources in the Earth’s crust [1,2,3]. So, sodium ion batteries have recently gained attention because the abundance of Na is huge and because these batteries are extremely favorable for large-scale stationary electric energy storage applications for renewable energy and smart grids [4,5,6]. Na layered transition-metal oxides Na*_x_*MnO_2_ (NMO) have been focused on as cathode materials of Na-ion batteries due to their similar structure to LiCoO_2_, for example Na*_x_*MnO_2_ [7], Na_x_CoO_2_ [8] and NaFeO_2_ [9]. Original investigation has shown that there is 0.15–0.22 Na reversibly extracted/re-inserted in α/β-NaMnO_2_ [10]. Ma et al. found that α-NaMnO_2_ can realize the first discharge capacity of 185 mAh/g at a rate of 0.1 C and a capacity of 132 mAh/g after 20 cycles [7]. Furthermore, β-NaMnO_2_ [11] also exhibited a first discharge capacity as high as 190 mAh/g, corresponding to the reinsertion of 0.82 Na per formula unit, and a discharge capacity of ca. 130 mAh/g after 100 cycles. An additional 160 mAh/g reversible capacity for Na_2/3_Ni_1/3_Mn_2/3_O_2_ cathode material can be obtained [12]. Layered Na_0.67_[Mn_0.65_Co_0.2_Ni_0.15_]O_2_ has also exhibited 141 mAh/g reversible capacity [13].

Nevertheless, Na*_x_*MO_2_ compounds exhibit low specific capacity, poor cycling performance, and high cutoff voltage. Na_0.7_CoO_2_ can deliver a charge capacity of 45.8–146.8 mAh/g when charged to 3.5V–4.8 V [14]. The insertion and extraction of Na ions will degrade the structure, and the active material becomes amorphous after several cycles [15]. Besides alternative element doping [12,13], surface modification has been found to effectively improve the performance of electrode materials [16,17]. A metal oxide (ZrO_2_ [18], ZnO [19], Al_2_O_3_ [17,20]) can be used as a coating layer to improve the structural stability between the active material and electrolyte. Al_2_O_3_-coated LiNi_0.5_Co_0.2_Mn_0.3_O_2_ shows a higher capacity retention of 85% than the pristine electrode (only 75%) after 100 cycles [17]. Metal oxides can effectively prevent contact between the electrode surface and the organic electrolyte, alleviating electrolyte decomposition. LiAlO_2_-coated LiNi_0.6_Co_0.2_Mn_0.2_O_2_ can maintain a reversible capacity of more than 149 mAh/g after 350 cycles [20]. In general, copper oxides are used as coating layers. For example, after a cycling test, a Cu_2_O-coated Si anode has less cracks than a pristine Si anode [21]. Dong et al. [22] demonstrated that NNMO@CuO cathode material combines the advantage of CuO coating and Cu^2+^ doping, and shows prominent cycling performance and rate capability. The CuO coating layer can prevent direct contact between the cathode active material and the electrolyte, and can suppress a side reaction between the cathode material and the unwanted by-product hydrofluoric acid [23]. In this paper, we use the wet chemical method to treat synthesized NMO. By this treatment, a thin coating layer can be formed on the surface, and some oxide coating layer and impurities can be removed from the surface of the material to form a smooth surface. In this way, the cyclic and conductive properties of NMO can be effectively improved.

## 2. Experimental Details

Na*_x_*MnO_2_ (NMO) was prepared by solid state reaction. First, Na_2_CO_3_ and Mn_2_O_3_ were weighed according to the molar ratio 1:1, and then the mixture was slightly ground. After that, ball grinding (ball material ratio 20:1) was used in an argon environment for 1h at the speed of 500 r/min. The mixed powder was put into a tubular furnace and heated to 500 °C at a heating rate of 4 °C/min for 2 h, and then was sintered at 700 °C for 8 h. Finally, the product was quenched to room temperature.

The quantitative Cu(NO_3_)_2_ calculated by stoichiometric ratio was added to distilled water to form a solution. Then, the obtained product was added into a blender operating at a speed of 200 r/min. The solution was continuously stirred and evaporated at 60 °C. The resulting powder was rapidly heated to 650 °C for 10 h at 10 °C/min using a tube furnace, and then was rapidly cooled to room temperature. By this wet chemical method, the copper oxide coating layer was obtained, and the partial copper oxide coating layer was removed to produce a smoothed surface. 

Untreated NMO or treated NMO was mixed with a conductive agent (acetylene black) and a binder (polyvinylidene fluoride) at a ratio of 8:1:1, and then a diluent (N-methylpyrrolidone) was added and the mixture was ground into a slurry. The obtained slurry was coated on an aluminum sheet, then placed in a vacuum oven (Shanghai, China) and dried at 90 °C for 12 h. After drying, the amount of active material adhered to each of the pole pieces was 1.5 to 2.5 mg/cm^2^. The electrolyte was composed of 1 M NaClO_4_ dissolved in the mixture of ethyl carbonate (EC) and dimethyl carbonate (DMC) (1:1 by volume) with 5% fluoroethylene carbonate (FEC). A sodium foil was used as a reference electrode, the prepared electrode was used as a working electrode, and the porous polypropylene was used as a separator to assemble a CR3023 type button battery. The discharged and charged performances were obtained by using the Land battery tester system (GAMRY, Wuhan, China). Electrochemical impedance spectroscopy (EIS) and cyclic voltammetry (CV) were measured on CHI 760E (Shanghai, China).

The X-ray powder diffraction (XRD) measurements were performed on a Bruker D8 ADVANCE diffractometer (Cu Kα, λ = 0.15406 nm) to determine the phase contained within the resulting material. The morphology of samples before and after treatment was characterized by field emission scanning electron microscopy (FESEM, Merlin, Zeiss) and high-resolution transmission electron microscopy (HRTEM, Titan G260-300). X-ray photoelectron spectra (XPS) were conducted on a Thermo Fisher Scientific K-Alpha+ using C 1s (B.E. = 284.8 eV) as a reference for observing the microstructure of the sample surface and the distribution of test elements.

## 3. Results and Discussion

According to the differential thermal analysis (DTA) shown in Figure 1a, it can be seen that the reaction is optimal when the temperature is higher than 650 °C. Then, the materials are sintered in the oxygen atmosphere under 650, 700, 750 and 850 °C, respectively, as shown in Figure 1b. It can be seen that along with the temperature increasing, the peaks gradually increase. According to the advantages and disadvantages of the assembled batteries, the optimal compound Na*_x_*MnO_2_ prepared by sintering at 700 °C is selected from the obtained compounds. The obtained product was identified by an X-ray powder diffractometer. The results are shown in Figure 1c. According to the comparison of the PDF card, the main peak is substantially identical to the map of Na_0.7_MnO_2.05_ (ICDD: 00-027-0752), and the space group is P63/mmc. The treated NMO was also passed through an X-ray powder diffractometer, and was compared with the NMO before the treatment (as shown in Figure 1d). The inset figure in Figure 1d shows enlarged XRD patterns of the pristine and treated samples between 13 and 19 °C. It can be seen that the low angle peak has shifted to the left, which is in agreement with other experimental results [22]. 

Copper oxide coating layers have been formed in treated NMO compounds. The peak shift could be because Cu^2+^ ions enter the host as dopant ions, and the ionic radius of Cu^2+^ (0.73Å) is larger than that of Na^+^ (0.69Å) and Mn^3+^ (0.72Å). 

In order to observe the morphology and structure of compounds, a scanning electron microscope (SEM) and a high-resolution transmission electron microscope (HRTEM) were used; the results are shown in Figure 2. According to Figure 2a,b, it can be seen that the obtained sample particles are granular in shape, and the edge portions are bonded together. The particle size is between 1.5 and 2 μm, and the whole particle is relatively uniform. Each particle has an obvious attachment, which makes the whole particle very rough (Figure 2b). In Figure 2e, it can be seen that the treated sample is obviously smoother than the untreated sample. The particle size is about 2 μm and is more uniform than the untreated sample. It can be clearly seen that there is almost no attachment on the surface (see Figure 2f). Hence, the sample surface after the treatment is smoother than that of the untreated sample. The particle size is slightly enlarged, and the overall particles are more uniform.

The morphology of the particles after the treatment was observed under high-resolution transmission electron microscopy (HRTEM). It can be observed that some particles have a coating layer of about 15 nm (as shown in Figure 2c,d). Figure 2g also shows that there is no obvious coating layer on the surface of some particles; a partial coating layer can be removed by washing. Figure 2h is the distance of (001) crystal plane with d = 0.519 nm. The treated NMO particles can be cleaned to remove part of the attachment. 

The EDS (Energy Dispersive X-Ray Spectroscopy) surface sweep (shown in Figure 2i,j) is made up of four elements of Na, Mn, Cu, and O on the NMO particles after the treatment. It can be seen that four elements are attached to the particles. However, the Na and O elements are not so uniform relative to other elements, and only a part of the area is attached with more Na and O elements. Combining with the line sweep of EDS in Figure 2k, the content of Na and O gradually increases in the direction indicated by the line, and it can be inferred that the surface of the pristine particle does adhere to a large amount of sodium oxide. After treatment, the deposit is significantly reduced, and Cu ions could also be introduced at the same time. It is important to improve the electronic conductivity and the performance of the entire battery after assembly into a battery.

In order to more intuitively observe the performance of the treated material, all electrodes are subjected to constant current cycling to study the cycle and rate performance from 2 to 3.6 V. Compared with untreated NMO, treated NMO has a significantly improved rate performance between 50 and 1000 mA/g. From Figure 3a,c, untreated NMO is 70.3, 51, 41.6, 32.2, and 22.5 mAh/g at 50, 100, 250, 500, and 1000 mA/g, respectively. The treated NMO is 123, 106.2, 94.4, 79, and 62.5 mAh/g, respectively. After the cycle test is completed, when the current returns to 50 mA/g, the specific capacities of NMO before and after treatment return to 61.2 and 122.1 mAh/g, respectively, which are 87% and 99.5% of the original discharge specific capacity. It is clearly better for the treated NMO in sodium ion intercalation/deintercalation reversible performance, which is in agreement with the NNMO@CuO sample, which also shows better performance [22]. The outstanding rate capability of a treated electrode can be ascribed to the smoothed surface that decreases barriers to improve Na-ion diffusion, and to enlarged layer distances when Cu^2+^ is doped into the crystal structure.

For the purpose of observing the degree of attenuation of the battery at the same current, batteries assembled with pristine NMO and treated NMO as cathode materials are run at the same current (50 mA/g) and 500 charge and discharge cycles. The data comparison is shown in Figure 3b; it can be clearly seen that the cycle performance of pristine NMO shows a very significant decrease in the first 50 cycles. The performance after the sample was treated also attenuates a part of the specific capacity, but the stability has a significant improvement. This might be because the structure of the electrode material itself, which is very thick and thereby does not provide many landing points nor much space for Na ion landing, and does not provide enough tunnel to shuttle. So, surface oxides make the insertion and deintercalation of Na ions more obstructive. The surface can be smoothed after wet chemical treatment, which allows Na ions to overcome such small resistance embedding and deintercalation. Comparing Figure 3a,c, it can be seen that except for the lower Coulomb efficiency of the first circle, there is almost 100% Coulomb efficiency from the second lap, which is also a good explanation for the smoothed coating layer. The structure of the NMO material itself undergoes internal structural phase change during the first cycle of charge and discharge [13,23]. After 300 cycles, pristine NMO can only maintain the initial specific capacity of 58.7%. Treated NMO still has 72.6%, which provides a better cycle life.

The cyclic voltammetry test curve (CV) is obtained under the condition of 2–3.6 V. It can be seen from the curve comparison before and after treatment, shown in Figure 3d, that the area surrounded by the treated curve is obviously larger than that of the untreated under the same rate. It is also consistent with the larger capacity. The oxidation peak and the reduction peak correspond to each other, which illustrates that the reaction is reversible. The first circle has a different curve from the number of cycles thereafter. It has been proved that the structural phase transition of the first few circles of NMO materials does occur [13,24].

In addition, electrochemical impedance spectroscopy (EIS) measurements of treated materials before and after cycling can also be used to demonstrate the improvement in performance after processing (Figure 4a,b). For pristine NMO, the material has a higher internal resistance than treated material, and shows a large semicircle in the high frequency region. For treated NMO, after 50 charge and discharge cycles, it still has a lower resistance (as shown in Figure 4b). The Nyquist plot clearly forms a semicycle, which means charge transfer resistance and Warburg impedance [22]. Then, the treated NMO generates stable SEI (Solid Electrolyte Interface) films that are formed by exfoliation of the treated NMO electrode surface during prolonged cycling. A smoothed surface and a thin copper oxide layer can form an effective protective layer on the surface of the electrode. This ultimately facilitates the transfer of electrons during electrochemical cycling.

## 4. Conclusions

In this paper, pristine NMO samples were prepared under different temperatures, which were 650, 700, 750 and 850 °C, respectively. The most suitable temperature was 700 °C, which had XRD peaks that were substantially uniform with reported Na_0.7_MnO_2.05_. All our discussion of structure and all electrochemical measurements were based on the sample of 700 °C. In order to improve the performance of NMO compounds, the wet chemical method was used to treat the pristine NMO with Cu(NO_3_)_2_ and form a coating layer at the position of the original oxide. Some of the coating layer is peeled off, smoothing the surface of particles by washing. Therefore, treated NMO can maintain 72.6% of the specific capacity after 300 cycles of charging and discharging cycles, which is better than the 58.7% for untreated NMO materials. The ratio of capacity remaining rate is increased from an initial 87% to 99.5%. So, this wet chemical method for treating electrode materials is available to improve electrochemical performance. It can not only reduce the internal impedance, but can also prevent the formation of SEI film during the battery cycle.

## Figures and Tables

**Figure 1 nanomaterials-10-00246-f001:**
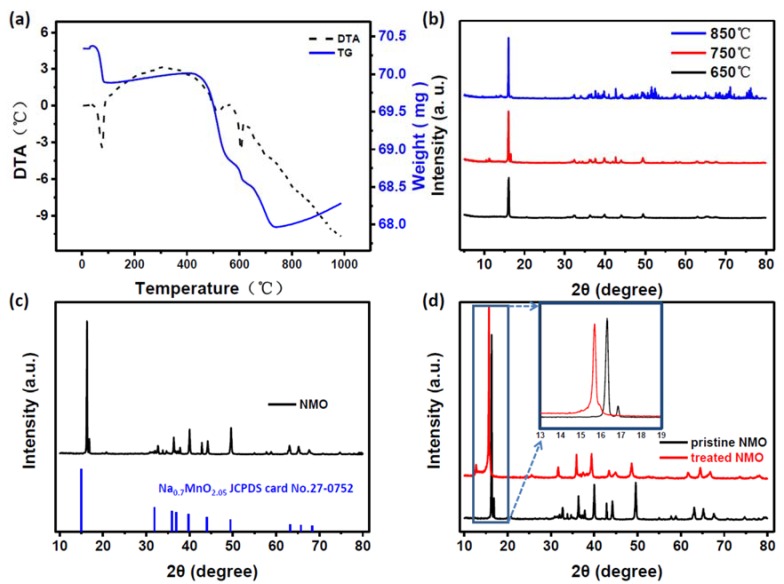
(**a**) Differential thermal analysis (DTA) results for NMO precursor mixtures; (**b**) X-ray powder diffraction analysis at different sintering temperatures; (**c**) comparison of NMO’s X-ray powder diffraction analysis results with the corresponding PDF card; (**d**) comparison of the results of X-ray powder diffraction analysis before and after treatment.

**Figure 2 nanomaterials-10-00246-f002:**
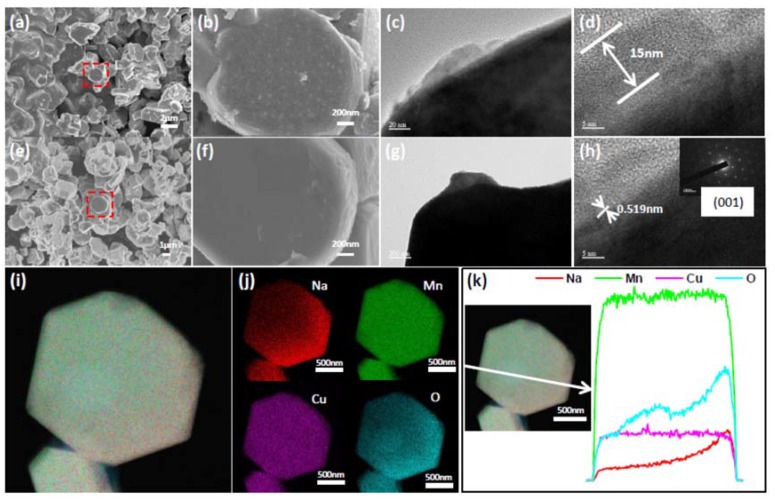
A series of characterization tests before and after treatment of NMO. (**a**,**b**) SEM results of NMO before and after treatment; (**e**,**f**) SEM results of treated NMO; (**c**,**d**,**g**,**h**) TEM results of treated NMO; (**i**,**j**) high-surface sweep mapping of treated NMO; (**k**) line scan data of treated NMO.

**Figure 3 nanomaterials-10-00246-f003:**
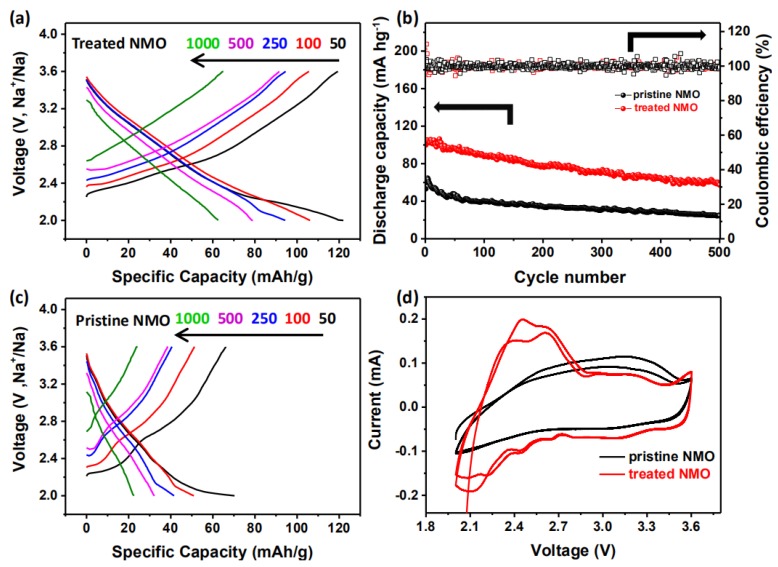
A series of electrochemical tests of NMO before and after treatment. (**a**), (**c**) Charge-discharge curves of NMO before and after treatment under different currents; (**b**) 500 cycles of charge and discharge of NMO before and after treatment at 50 mA/g current; (**d**) cyclic voltammetry curves of NMO before and after treatment.

**Figure 4 nanomaterials-10-00246-f004:**
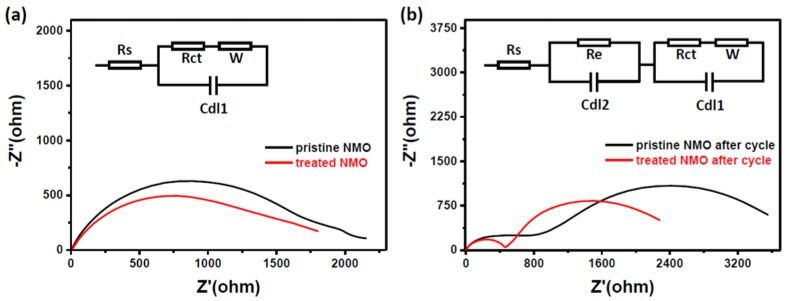
Electrochemical impedance spectroscopy (EIS) before (**a**) and after (**b**) charge and discharge, after assembly of pristine NMO and treated NMO into a battery.

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
