# Peer review of "Smoothing the Surface and Improving the Electrochemical Properties of NaxMnO2 by a Wet Chemical Method"

_nanomaterials, 2020, doi:10.3390/nano10020246_

Round 1

Reviewer 1 Report

The authors present an interesting article facing the challenges of Na based layered oxide cathode materials. Useful data and insights are provided. Revision is recommended and suggested to improve the quality.

Figure 3: I suggest to use similar scale for x-axis for figure 3 a) and b) for uniformity reasons and the fact that the beneficial effect of treated NMO would get more obvious.

In Figure d) voltage is used while in a) and c) “potential…”. Please explain the non-uniformity. As I understand both are coin cells, so I suggest this “voltage” term is the correct one, as the “potential” can only be measured in a three electrode setup.

In two electrode set up like in e.g. coin cells only the voltage can be measured. Please consider the following literature for you experimental:

Nölle, K. Beltrop, F. Holtstiege, J. Kasnatscheew, T. Placke, M. Winter, Materials Today, (2019).

I suggest “…The fabricated coin-cells were discharged/charged using Land battery tester system (Wuhan, China). Since it is a two electrode setup, the maeasured voltage is only an approximation to potential values.”

Results and discussion:

“And the performance after the treated sample also attenuates a part of the specific capacity, but the stability have a significant increase.”

Has, not have.

“This might be reason that the structure of th electrode material itself, which is but very thick,..”

The “but” is wrong I guess.

Figure4: The differences in resisitances/overpotentials are obvious from figure 3. Which additional information do you get from EIS?

Author Response

Dear editor, 

Firstly, we have been revised English for full paper; Secondly, according to reviewer’s questions, our reply is follow as:

1, Figure3(b and d) have been revised.

2, we have been read literature (Nölle, K. Beltrop, F. Holtstiege, J. Kasnatscheew, T. Placke, M. Winter, Materials Today, (2019)), and adjusted our presentation about experimental parts.

3, we have been presentated something that has proposed by reviewer in results and discussion parts.

Thanks!

Your sincerely!

Reviewer 2 Report

The paper deals with a surfce treatment for a manganese oxide, and it hsows that the treatment greatly improves electrochemical performance. The paper is well written and contains enough elements of novelty in my view to be published, upon addressing minor points:

Please check the spelling; Include relevant references about organic (polymeric) approaches to nanomaterials: 10.1021/acs.chemrev.8b00286; doi: 10.1039/C4OB02643H

Author Response

Dear editor,

we have been revised English for full paper. 

Thanks!

Your sincerely!

Round 2

Reviewer 1 Report

The authors revised the manuscript accordingly. It can be accepted in present form.